# Synthesis and Performance of a Novel Cotton Linter Based Cellulose Derivatives Dispersant for Coal–Water Slurries

**DOI:** 10.3390/polym14061103

**Published:** 2022-03-10

**Authors:** Chengli Ding, Xiao Zhu, Xue Ma, Hongsheng Yang

**Affiliations:** Key Laboratory of Coal Clean Conversion & Chemical Engineering Process, Xinjiang Uygur Autonomous Region, College of Chemical Engineering, Xinjiang University, Urumqi 830046, China; zhuxiao@stu.xju.edu.cn (X.Z.); maxue@stu.xju.edu.cn (X.M.); yanghongsheng@stu.xju.edu.cn (H.Y.)

**Keywords:** cotton linter cellulose, dispersant, coal–water slurry, stability

## Abstract

A novel sulfonic-cellulose succinate half ester (S-CSHE) dispersant for coal–water slurry (CWS) was successfully synthesized using cotton linters, sulfamic acid and succinic anhydride in DMF by a one-pot synthesis. The effects of the synthetic condition of S-CSHE as a dispersant for CWS were studied. An S-CSHE with a maximum degree of substitution of 0.98 was obtained under these optimized conditions. The synthesized samples were characterized by GPC, FT-IR spectroscopy, ^13^C-NMR, and SEM. The molecular weight was from 2.2 × 10^3^ to 1.2 × 10^4^, revealed by GPC; FT-IR spectra analysis revealed characteristic absorptions of the sulfonic-cellulose succinate half ester; SEM images showed smooth cellulose structures, while the S-CSHE had a compact surface. Effects of S-CSHE on the fluidity of application as the dispersants for the CWS prepared from Chinese Zhundong coal were studied further. The CWS application performance investigations showed that S-CSHE can most effectively reduce CWS viscosity, and perform excellent dispersity and stability. When the dosage of S-CSHE was up to 0.5 wt.%, the maximum coal content of CWS may reach 70 wt.%, and the apparent viscosity of CWS was 487 mPa·s. The CWS prepared using S-CSHE (0.38 wt.%) for Zhundong coal showed the rheological characteristics of shear-thinning, and is consistent with the Herschel–Bulkley model. This work found a new route for utilizing cotton linters cellulose and enlarged the selecting range of the dispersant for CWS. It has a positive significance for efficient and clean utilization of Xinjiang Zhundong coal.

## 1. Introduction

The efficient and clean utilization of coal resources is of significant interest in recent years. Coal gasification technologies, especially the opposed multi-burner (OMB) gasification technology using coal–water slurry (CWS) as a feedstock, can achieve the efficient and clean utilization of coal resources, which attracted the attention of the world [1]. The desirable CWS should have properties, such as a high coal content for economic consideration, excellent stability for storage, and a viscosity as low as possible for transportation and combustion. To achieve the demands, the selection of a dispersant plays an extremely important role [2,3]. Some dispersants have been commercialized, such as naphthalene sulfonated formaldehyde condensates, sulfonated melamine formaldehyde polymers and the aminosulfonic acid series, etc. They are in the process of synthesis or use may result in undesirable environmental effects, as they can accidentally or intentionally release formaldehyde into the environment [4,5,6,7,8]. Therefore, an environmentally friendly and biodegradable additive is urged to be developed as a new high-performance dispersant for coal–water slurries using cheap and abundant natural renewable resources as raw materials.

Cellulose is the main representative of such resources. Cellulose is one of the most widely distributed biopolymers on earth, having many unique advantages, such as low cost, renewability, being environmentally friendly, natural biodegradation, chemical stability and biocompatibility [9,10]. Cellulose-based materials have attracted much attention and have already been applied in food, water treatment, membrane materials, drug delivery, 3D printing and other fields [11,12,13,14,15]. In the last few years, research and technology of water-soluble cellulose derivatives as water-reducing agents of cement concrete have made remarkable discoveries in the field of bio-renewable materials [16,17]. Wang, L. J. have synthesized water-soluble sulfobutylated cellulose ether (SBC); the study results indicate that water-soluble SBCs have the potential to be developed as a set-retarding and water-reducing agent—even a high-range water-reducing agent [18]. However, to the best of our knowledge, there is no report on the direct synthesis of dispersants for coal–water slurries.

In this investigation, we have developed cellulose derivatives containing sulfonate and carboxylate esters through one-pot synthesis. This method utilizes sulfamic acid to obtain high degrees of sulfonate substitution (DS sulfonate), while sulfamic acid is used as a catalyst for the esterification reactions. The CWS application properties, such as viscosities and stabilities, were investigated in the presence of the dispersants. Based on the investigations, the feasibility of the described method provided a basis for the synthesis, characterization and application of cellulose-based dispersant.

## 2. Materials and Methods

### 2.1. Materials

Cotton linter was obtained from the Aksu Cotton Linter Processing Company (the content of cellulose in cotton linter was 94.34%) (Xinjiang, China). Sulfamic acid, urea and other reagents (analytical grade) were obtained from the Tianjin Guangfu Chemical Company (Tianjin, China).

Zhundong coal from Fukang, Xinjiang Province in China, was used in this study. Table 1 shows the results of the proximate analyses of coal.

The particle size distribution of the coal sample is given in Table 2.

### 2.2. Methods

The synthetic route for S-CSHEs included esterification by sulfamic acid and succinic anhydride, as shown in Figure 1.

The degrees of degradation, sulfonation and esterification of S-CSHE could be controlled stoichiometrically, and by adjusting the time and temperature of the chemical reaction. Sulfamic acids were used as reactants, while sulfamic acid also proved to be a cost-effective and recyclable catalyst for esterification. The catalytic application of sulfamic acid was recognized due to its easy separation, recyclability, greater economic viability and environmental friendliness [19,20,21,22,23,24].

Preparation of cellulose derivatives containing sulfonate and carboxylate groups were cotton linters cellulose (8.1 g, ~0.05 mol), sulfamic acid (13.55 g, ~0.15 mol), succinic anhydride (6 g, ~0.06 mol), urea (3.0 g, ~0.05 mol), and DMF, and were mixed in a 100 mL three-necked flask equipped with a thermometer. The mixture was warmed to 120 °C and was stirred for 6 h. The products were filtered, washed with ethyl alcohol, and dried at 80 °C under vacuum. Then, it was dissolved in distilled water, and the solution was dialyzed against distilled water. The final samples were obtained after being freeze-dried by using a freezer dryer at −50 °C for 2 days.

#### 2.2.1. Determination of Degree of Substitution (DS)

The extent of ester, as defined by the succiniclic acid content (O-C=O %) and the sulfur content (S%), was measured using an X-ray photoelectron spectroscopy (D/max2500, Bruker AXS, Beijing, China). The DS was calculated according to Equation (1) and Equation (2), as follows [8,25]:(1)$$DS_{s}=\frac{162\omega_{s}}{44-122\omega_{s}}$$
(2)$$DS_{c}=\frac{162\omega_{c}}{32-102\omega_{c}}$$
where ω_s_ and ω_c_ are the mass fraction of the sulfonic and carboxyl groups; 122, 102 is the molar quantity of the substituted group; 162 is the molar quantity of a grape unit; 44 is the molar quantity of the O-C=O; 32 is the molar quantity of the S.

#### 2.2.2. Fourier Transform-Infrared (FT-IR) Spectroscopy

The chemical structures of cellulose and sulfonic-cellulose succinate half ester (S-CSHE) were characterized by FT-IR spectroscopy (FT/IR-430, JASCO, Tokyo, Japan). The samples were prepared by grinding to a fine powder (sample/KBr 1:1000 ratio). FT-IR spectra were recorded over the wavenumber range from 500 cm^−1^ to 4000 cm^−1^.

#### 2.2.3. Nuclear Magnetic Resonance Spectroscopy (NMR)

^13^C-NMR spectra of S-CSHE samples were recorded in D_2_O (50 mg/mL) containing tetramethylsilane as an internal standard. The spectra were acquired on an Inova-400 m superconducting nuclear magnetic resonance spectrometer(Bruker, Billerica, MI, USA) at room temperature, collecting 16 scans.

#### 2.2.4. Scanning Electron Microscopy (SEM)

The morphologies of the samples were imaged with field emission scanning electron microscopy (FESEM, Hitachi SU800 equipped with an EDS system) (Garl Zeiss AG, Oberkochen, Germany), in which all samples were spread on conductive carbon tape and sputtered with gold at an acceleration voltage of 10 kV.

#### 2.2.5. Molecular Weight Distribution

The samples were dissolved in high-purity water with a concentration of 8.00–10.00 mg/mL at the appropriate temperature; prior to injection, the sample solutions were filtered. Following the principle of GPC, the molecular weight distributions of the samples were estimated in Table 3.

#### 2.2.6. Viscosity Measurement and Rheological Property Determination

The viscosities of CWS were measured by using a rheometer (AR-2000, TA Instrument Company, New Castle, DE, USA). The shear rate range was 0–100 s^−1^, and the temperature was maintained at 25 °C.

#### 2.2.7. Adsorption Amount Measurement

The coal–water slurry was prepared with 10 wt.% of coal and the known concentration of a dispersant. The slurry was stirred at 1200× *g* rpm for 10 min, then centrifuged, and the supernatant was used for determining dispersant equilibrium concentration, employing a UV–vis spectrophotometer (UV-265FW, Shimadzu Corp., Tokyo, Japan). The concentration of the dispersant in the solution was determined from the characteristic absorption wavelength according to a predetermined calibration curve. The adsorption amount was calculated as follows [8,25]:(3)$$\Gamma_{i}=\frac{\left(C_{0i}-C_{ti}\right)V}{m}$$
where $\Gamma_{i}$ is the adsorption amount per unit mass coal (mg·g^−1^), $C_{0i}$ and $C_{ti}$ are the initial and final mass concentrations of the dispersant (mg·L^−1^), V is the total volume of the solution (L) and m is the mass of the coal sample (g).

## 3. Results and Discussion

### 3.1. FT-IR Analyses

The IR spectra of the cotton linters cellulose and S-CSHE with DS = 0.98 are shown in Figure 1.

The FT-IR of the S-CSHE were in good agreement with those reported in the literature [26]. The peak at 1722 cm^−1^ or 1618 cm^−1^ assigned to O=C was clearly present [27]. The strong peak at 1225 cm^−1^ was visible in the spectra of S-CSHE only, and corresponded to the symmetrical vibrations of S=O, while the peak at 807 cm^−1^ was attributed to symmetrical vibrations of S-O [28]. These results indicated that both the sulfonate and carboxyl groups were introduced into the cellulose molecules.

### 3.2. Morphological Analysis by SEM

The SEM micrographs of the cotton linters cellulose and the S-CSHE are shown in Figure 2.

In this study, we examined the microstructures of the samples in order to understand the effect of esterification on the cellulose structure shape and particle size. Through the visible analysis of cotton linters cellulose and S-CSHE SEM, the surface of S-CSHE was denser than that of cotton linters cellulose, indicating that there were no fragments to the cotton linters cellulose and its morphology had been completely destroyed during the sulfonate and carboxylate modification. Esterification of the surface hydroxy groups of cellulose with sulfonic and carboxy groups changed the original morphology.

### 3.3. ^13^C-NMR Characterization

As shown in Figure 3, the absorption peak at ca. 70.28 ppm could be assigned to the substituted C6 position (C-6S) due to the low field shift [29], which may result from the sulfonic group substitution of the active hydroxyl group of the cellulose; and the absorption peak at ca. 82.00 ppm was ascribed to the substituted C2 position (C-2S). This could probably be explained by the strong shielding effect induced by the esterification of the hydroxyl groups in the cellulose structure, which may lead to the esterification of the hydroxyl groups of AGU, resulting in the displacement of the carbon resonance nearby toward the high field [30]. The peak which belongs to the substituted C6 position (C-6S’) was observed near 58.14 ppm. Two signals around 68.77 ppm and 66.74 ppm were found for C3S, C3S’, respectively, indicating the existence of the substituted C3 position. An observation of the chemical shift of C-6S, C-2S, C-3S in the ^13^C-NMR spectrum of S-CSHE indicated the substitution of the hydroxyl group in cotton linters cellulose.

### 3.4. Viscosity and Rheological Characteristics of CWS

Viscosity is one of the most important rheological characteristics of CWS. In order to evaluate the viscosity-reducing capacities of the different substitution dispersants, the apparent viscosities of the CWSs with a coal content of 65 wt.%, neutral pH and different dispersant dosages are exhibited in Figure 4.

As shown in the figure, with the different degrees of substitution (DS) of the sulfonic-cellulose succinate half ester (S-CSHE), the apparent viscosity decreases abruptly with increasing dispersant dosage, reaches a minimum and then increases above a certain dosage. The minimum viscosity corresponds to the optimal dispersant dosage (0.51 wt.%). Three factors could account for the viscosity increase above the optimal dispersant dosage. Firstly, the increase in the counter-ion density (Na^+^) will compress electric double layers, reducing the relatively long-range electrostatic repulsive force. Secondly, the structure of the adsorbed polymer dispersant on the particle surface will be compressed by the negative charge (-COO^−^, SO_3_^−^) in the sulfonic-cellulose succinate half ester (S-CSHE), which reduces the short-range steric repulsive force between particles. Both reducing mechanisms increase the suspension viscosity [31,32]. Thirdly, the impact of the differences in molecular weight (M_W_) and the degree of substitution (DS) on the rheological behavior was caused [33].

It could be concluded that DS = 0.75 has the best viscosity-reducing capacity among the three dispersants at the optimal dispersant dosage, which may be attributed to the molecular weight. At the dosage, the apparent viscosity of the CWS prepared from DS = 0.75 gives a minimum of around 461 mPa·s. Additionally, the apparent viscosity of the CWS prepared from DS = 0.84 is less than that of the CWS prepared from DS = 0.98 at 0.48 wt.% dispersant dosage, which may be due to the larger steric repulsive force between particles that resulted from the flexible structure of the DS = 0.84 molecule.

It is well known that CWS with shear-thinning rheological behavior is the desired result in industries. Figure 5 shows that the viscosities of CWS samples were relatively high at a low shear rate, and apparent viscosities sharply decreased with the increase in the shear rate, which is consistent with the shear-thinning characteristic. It fully meets the requirement that the apparent viscosity of industrial CWS should be less than 1200 mPa·s when the shear rate is 100 s^−1^. The viscosity OF S-CSHE that compared with naphthalene sulfonic acid formaldehyde condensates (NSF) of earlier reported similar research, is consistent with the literature reports [34,35].

The rheological properties of CWS indicate that the flow characteristics of CWS are in accordance with the Herschel–Bulkley model ($\tau =\tau_{0}+k\gamma^{n}$) [8,36]. The fitting results of the experimental data are shown in Figure 6 and Table 4.
$$\tau =\tau_{0}+k\gamma^{n}$$
where τ is the shear stress, τ_0_ is the yield stress, k is the fluid consistency index, γ is the shear rate and n is the flow behaviour index.

### 3.5. Static Stability of CWS

Stability is an important index for the quality of CWS. In this experiment, the slurry samples were allowed to stand still for 7 days, and then the water separation ratio (WSR) was measured to appraise the static stability of CWS (Table 5).

As seen in the table, the penetration ratio of the CWS prepared from each one of the three dispersants decreases with the increasing coal content, but the CWS prepared from DS = 0.75 has the highest penetration ratio at the same coal content, which indicates the best ability of DS = 0.75 in stabilizing CWS. This might be due to the flexible chain structure of DS = 0.75 molecule. Moreover, for practical application, CWS should have a low viscosity and to give good stability, depending on the dispersant concentration, coal type, particle size distribution of coal, coal concentration, etc. [37,38].

### 3.6. Adsorption and Dispersion Mechanism of S-CSHE

According to the characteristics of cellulose and its modifiers discussions, the adsorption and dispersion mechanism of S-CSHE in the CWS preparation is speculated, as displayed in Figure 7.

The middle-rank coal of the Zhundong area of Xinjiang has contained pore structure, high internal water content, and oxygen-containing side-chain functional groups. According to the principle of similar connection, cellulose structural chain can be well combined with middle-rank coal side-chain and oxygen-containing functional groups. The long molecular chain and strong electrostatic repulsion play an effective role in dispersion and stability [39].

The long-chain structure of dispersant can stabilize the surface of coal particles and make it difficult for the particles to coagulate, thus improving the dispersion effect. The introduction of carboxyl and sulfonic groups of cellulose can not only connect each monomer closely, but can also serve as the hydrophilic functional group in the dispersant system. This increases the hydrophilic ability of dispersant and resistance to metal ions in coal, making it easier to form a stable hydration film and thus, improving the dispersion stability [8,40].

## 4. Conclusions

Based upon this study, S-CSHEs can be developed as a new dispersant for coal–water slurries by means of the chemical modification of cotton linters cellulose using C_4_H_4_O_3_-NH₂SO_3_H-DMF as a modifying agent by a one-pot synthesis. The results demonstrated that the DS is crucial to control the fluidity, stability and viscosity of CWS paste. It was noted that, as DS increased, the molecular weight was diminished. Furthermore, the flexibility of the S-CSHE molecule and the roughness of the molecular surface became smaller. It could be concluded that S-CSHE (DS = 0.75) had a better viscosity-reducing ability and dispersibility, as well as stabilizing ability for CWS. The improvements in the application performance were mainly attributed to the special cellulose structure. The CWS prepared using S-CSHE (0.38 wt.%, DS = 0.75) for Zhundong coal showed the rheological characteristics of shear-thinning, and were consistent with the Herschel–Bulkley model. Meanwhile, this research is a new way to enhance the utilization value of the renewable cotton linter of Xinjiang and widens the selection range of the dispersant of CWS.

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
