# Peer review of "Synthesis and Performance of a Novel Cotton Linter Based Cellulose Derivatives Dispersant for Coal–Water Slurries"

_polymers, 2022, doi:10.3390/polym14061103_

Round 1
Reviewer 1 Report
This paper reports a novel one-pot synthesis method for sulfonic-cellulose succinate half ester (S-CSHE) using cotton linters cellulose, sulfamic acid, and succinic anhydride in DMF. They characterized S-CSHE with GPC, FT-IR, 13CNMR, and SEM. They also evaluated the applicability of S-CSHE as dispersant for the coal–water slurries (CWSs) prepared from Chinese zhundong coal.
The significance of this paper is that it reports S-CHSE as an environment friendly and biodegradable alternative to anionic polymer dispersant for CWS. The authors find that adding S-CSHE reduces the viscosity and improves the dispersibility and stability of CWS.
The study is technically sound, and the results are well discussed. The authors also provide a good background and introduction to the study. No further experiments are needed, but I think this manuscript could be strengthened by the suggestions listed below:
- Line 32-34: Can the authors provide some references for the statement?
- Line 54-57: The authors report S-CSHE as an alternative to the toxic anionic polymer dispersant for CWS. It might be helpful to know how the viscosity reducing capability and stabilizing ability of S-CSHE compares with those of anionic polymer dispersants.
- It might be helpful to provide references for equations 1, 2, and 3.
- Line 171-176: The authors report SEM images showing that the smooth morphology of cellulose being disrupted in S-CSHE due to sulfonate and carboxylate modification. Is there a hypothesis how this change in morphology might affect the end functionalities of the S-CSHE?
- In Figure 4, the apparent viscosities are reported for a shear rate of 100 s-1. Can the authors elaborate a bit on why this shear rate was chosen for the study?
- Line 238: Is the penetration ratio same as the water separation ratio?
- Line 241: It is mentioned that DS=0.75 has a flexible chain structure. It might be useful if the authors mention how the chain flexibility was measured/calculated and how it compared to the other DS.
- In the conclusion, the authors comment on the degree of crystallinity and contact angle. They, however, have not been discussed in the main text. It will be more helpful to relate these parameters to the study if the authors mention how their statements on crystallinity and contact angle were derived.

Author Response
Please find our response to reviewer's comments in the attachment.

Reviewer 2 Report
The paper includes Table 1. and Figures 1 to 4 and also 7 not discussed in the paper. What purpose did the authors have in presenting them in the paper without discussing them? If they are unnecessary please remove them or discuss the presented elements.
The literature review lacks information on the types of biomass that can be sources of cellulose and its content in the different types of biomass.
DOI:10.5604/01.3001.0014.5531
DOI:10.26202/sylwan.2019136
DOI:10.1007/s00226-021-01350-1
Please check scheme 1.
DOI:10.1016/j.cattod.2013.09.022
DOI: 10.1039/C8RA05983G
doi: 10.1002/cssc.201000181
https://doi.org/10.18052/www.scipress.com/ILCPA.57.72
Author Response

(The authors gave the same response as above.)

Reviewer 3 Report
Interesting results and novelty work. A paper focuses on Synthesis and performance of a novel cotton Linter based cellulose derivatives dispersant for coal–water slurries. Though the intention of the authors is highly commendable, there is lot of problems particularly in the presentation throughout the manuscript. Besides there are many grammatical mistakes throughout the manuscript, particularly in respect of use of singular and plural with the subject or verb. In view of the above comments, whole manuscript should be properly written to make it acceptable by Polymers journal. I highly recommended this article to be accepted and published in the revised version.
Abstract:
The abstract given here starts without any background for the present work. Of course, it contains brief details about experimental aspects and the obtained results. However this abstract does not follow the norm of an abstract, which should state briefly:
- The purpose of the study undertaken, what are you trying to solve
- brief mention of experimental aspects (without using abbreviations)
- highlights of the results numerically
- Important conclusions based on the obtained results
- Potential applications
Therefore, it is suggested that the Abstract to be modified as per the suggestions given above.
Introduction
Introduction section is long with a many references based on the literature survey conducted by the authors. This is very good. However, this lacks in proper presentation of literature survey, which should have been systematic whereby existing scientific gaps should have been brought out. This should have given justification for the present study, which should be followed by the objectives of this study. In fact there is large amount of literature available on the characterization of cotton linter based cellulose . Similarly, a large number of methods to obtain these materials have been used mentioning their advantages and limitations. None of these have been brought out in this study whereby the authors have not justified why they have chosen the method they have used in their study. It should be noted that normally 'Introduction' should give brief background through literature survey for the study citing previous published work where-by scientific gaps that exist should be brought out. This would have led to justification for the present study. It is therefore suggested that ‘Introduction Section’ should be revised as suggested above because this Section is an important one from the point of view of taking up the present study.
Relevant article on lignocellulosic should be cited such:
Polymers (Basel) 2020;13:116. https://doi.org/10.3390/polym13010116.
Polymers (Basel) 2021;13:2710. https://doi.org/10.3390/polym13162710.
Polymers (Basel) 2021;13:1–51. https://doi.org/10.3390/polym13193365.
Carbohydr Polym 2018;181:1038–51. https://doi.org/10.1016/j.carbpol.2017.11.045.
J Mater Res Technol 2019. https://doi.org/10.1016/j.jmrt.2019.08.028.
Int J Biol Macromol 2019;123:379–88. https://doi.org/10.1016/j.ijbiomac.2018.11.124.
Food Hydrocoll 2020;98:105266. https://doi.org/10.1016/j.foodhyd.2019.105266.
J Mater Res Technol 2019;8:4819–30. https://doi.org/10.1016/j.jmrt.2019.08.028.
BioResources 2017;12:8734–54. https://doi.org/10.15376/biores.12.4.8734-8754.
In my opinion the paper will have good merit if such applications can be demonstrated and reported. Can you give some example?
Materials and Methods:
Normally, this section should have two main subsections. The first one is Materials which should give details of all materials used in the study, where from they were procured, known characteristics, if available (for e.g. cotton linter, cement, coal, where do you get it, what is the purity of the chemical and etc.).
The second subsection should be Methods, where methodologies used in the study should be given in a systematic way using sub section with numbers for each of the properties. First the processing or preparation aspects of the final material should be given followed by the characterization of prepared materials including preparation of samples for any specific property or morphology studies should be presented in a systematic way. Here one should also clearly mention the number of samples used, any standards followed for variety of properties, make and model of the instruments used for characterization, their accuracy and experimental conditions used, etc.
It should be known to the authors when one publishes any scientific paper, the results presented therein should be such they should be reproducible by any other person when the experiment is repeated using the same materials. In the present paper, it would be difficult for any other person to repeat the experiments because the chosen materials do not have any pre-history, which is required for other researchers to carryout experiments to check the possible reproducibility of the procedure adopted by these authors.
Some of the paragraph should be under results and discussion and if it is already there then this becomes repetition and hence can be deleted. Secondly, this Section is methods and hence only results should be mentioned and then it should be discussed preferably comparing it with earlier reported similar results by other researchers.
It is better to do some experiment on crystallinity XRD. In my opinion the paper will have good merit if such properties can be demonstrated and reported as it shows fully potential of the starch.
Results & Discussion
Well written and easy for the reader to understand what the authors have conveyed.
Some of the paragraph should be under Methods and if it is already there then this becomes repetition and hence can be deleted. Secondly, this Section is Results & Discussion and hence only results should be mentioned and then it should be discussed preferably comparing it with earlier reported similar results by other researchers.
Throughout the manuscript, there are less comparison had been done with other published journal. Therefore, please support your statements with other researcher’s work in the section result and discussion. It should be discussed preferably comparing it with earlier reported similar results by other researchers.
How many sample did for each experiment? Please do ANNOVA test and standard deviation for all data collected and presented.
Scheme 1, Figure 1, Figure 3, Figure 4, Figure 5 is not clear. Please revise.
Please label Figure 2
Conclusions
Conclusions given here are do not reflect what had been achieved including many speculations. It is too long and should be in 1 paragraph. Hence these need to be suitably modified. It may be remembered that this Section forms a summary of all the major observations/ results obtained. Accordingly, here presentation should consist of the main Results or the observations of the study in short sentences probably with bullet points. This should stand alone or form a subsection of a Discussion or Results Section. Hence better to rewrite this Section based on the comments given in the whole text.
General Comments:
The paper though contains some interesting results and novelty work, it lacks in its proper presentation in the whole manuscript. Of course there is need for better language throughout the manuscript. It is suggested that the authors should take the help of native English speaking person to take care of this problem. In view of these, the paper is highly recommended and should be accepted for publication in the revised form. It is suggested that the authors should revise the paper in the light of above comments/suggestions.
Author Response

(The authors gave the same response as above.)

Round 2
Reviewer 3 Report
Please answer one by one the comments given by the reviewer.
Please revise back.
Author Response
Dear reviewer
Greetings!
Thank you very much for giving us an opportunity to revise our manuscript. We appreciate you very much for their constructive comments and suggestions on our manuscript entitled.
Best regards
chengli Ding

Round 3
Reviewer 3 Report
The authors had revised the manuscript according to the reviewer's comments. Thus this manuscript should be accepted in its current form.